# Gut Dysbiosis and the Intestinal Microbiome: *Streptococcus thermophilus* a Key Probiotic for Reducing Uremia

**DOI:** 10.3390/microorganisms7080228

**Published:** 2019-07-31

**Authors:** Luis Vitetta, Hannah Llewellyn, Debbie Oldfield

**Affiliations:** 1Sydney Medical School, Faculty of Medicine and Health, The University of Sydney, Sydney NSW 2006, Australia; 2Medlab Clinical, Sydney NSW 2015, Australia

**Keywords:** *Streptococcus thermophilus*, lactic acid bacteria, chronic kidney disease, uremic toxins, probiotic treatments, inflammation, mucosal immunity

## Abstract

In the intestines, probiotics can produce antagonistic effects such as antibiotic–like compounds, bactericidal proteins such as bacteriocins, and encourage the production of metabolic end products that may assist in preventing infections from various pathobionts (capable of pathogenic activity) microbes. Metabolites produced by intestinal bacteria and the adoptions of molecular methods to cross-examine and describe the human microbiome have refreshed interest in the discipline of nephology. As such, the adjunctive administration of probiotics for the treatment of chronic kidney disease (CKD) posits that certain probiotic bacteria can reduce the intestinal burden of uremic toxins. Uremic toxins eventuate from the over manifestation of glucotoxicity and lipotoxicity, increased activity of the hexosamine and polyol biochemical and synthetic pathways. The accumulation of advanced glycation end products that have been regularly associated with a dysbiotic colonic microbiome drives the overproduction of uremic toxins in the colon and the consequent local pro-inflammatory processes. Intestinal dysbiosis associated with significant shifts in abundance and diversity of intestinal bacteria with a resultant and maintained uremia promoting an uncontrolled mucosal pro-inflammatory state. In this narrative review we further address the efficacy of probiotics and highlighted in part the probiotic bacterium *Streptococcus thermophilus* as an important modulator of uremic toxins in the gut of patients diagnosed with chronic kidney disease. In conjunction with prudent nutritional practices it may be possible to prevent the progression of CKD and significantly downregulate mucosal pro-inflammatory activity with the administration of probiotics that contain *S. thermophilus*.

## 1. Introduction

The intestinal microbiome is subject to daily perturbation that most probably occur with daily environmental/dietary and psychological/physical stressors as does the administration of antibiotics [1,2,3]. The intestinal microbiome, therefore, presents a complex and highly individualized picture of bacterial taxa diversity and abundance that can be subject to continual low-level disruption that in disease states may lead to disease progression [1,4].

Chronic kidney disease (CKD) characterizes a continuum of chronic changes in the kidney that includes non-specific tissue changes of chronic progression with chronic inflammation, tubulointerstitial fibrosis, glomerulosclerosis, and vascular rarefaction [5,6]. The accepted classification of the different stages of chronic kidney disease describes the progression of uremic illness that can be related to an accumulation of gut derived toxins.

Local and systemic inflammatory processes that are supported by the overproduction of uremic toxins track a dysbiotic bacterial cohort that typically occupies the proximal colonic lumen (i.e., ascending colon) [7]. In patients with progressive CKD, uremic toxins such as p-cresyl and indoxyl sulphates and blood bound proteins that are poorly cleared by dialysis, are constantly associated with increased severity of CKD [8]. Therefore, uremic toxins have been postulated to exert harmful effects via augmentation of glucotoxicity and lipotoxicity, increased activity of the hexosamine and polyol biosynthetic/biochemical pathways, and the subsequent buildup of advanced glycation end products that are associated with a uremic dysbiotic colonic lumen [7]. The colonic environment is consequently altered through the active accumulation of numerous endogenous microbial metabolites that include nitrogenous compounds such as oxalic acid, uric acid, and urea [9]. These intestinal bacteria-derived nitrogenous toxic metabolites that are normally excreted by the kidneys are transported across the intestinal epithelia and subsequently into the systemic circulation where they are retained [9].

### What this Narrative Review Proposes to Add

The view that the intestinal tract was a collection of toxic waste and that it was largely an inert organ has been significantly challenged over the last decade. This review presents an overview of the laboratory and in vivo research that has demonstrated that the intestinal microbiota provide essential metabolic and physiological functions for human survival such as the harvesting of essential food nutrients, vitamins, and energy; metabolism of xenobiotics; protection from opportunistic pathobionts (pathogenic capabilities); influence neurological pathways; development and maturation of the intestinal epithelium; and the development and maintenance of homeostasis of localized immune function. In this respect and in a neonate, the importance of microbial metabolic activity is evidenced via the assault of all mucosal surfaces and the skin where the bacteria that colonize the gastrointestinal tract provide essential cues for the development of immunological tolerance and metabolic homeostasis over a lifetime for the host.

Lactic acid producing bacteria from the genera *Lactobacillus* and *Bifidobacteria* as well as *Streptococcus thermophilus* have demonstrated beneficial health effects in humans. *Streptococcus thermophilus* is a Gram-positive bacterium belonging to the phylum Firmicutes, family Streptococcaceae, and order Lactobacillales. *Streptococcus thermophilus* belongs to the clade of Lactic acid producing bacteria, that includes species from the genera *Carnobacterium, Enterococcus, Lactobacillus, Lactococcus, Leunostoc, Oenococcus*, *Pediococcus, Tetragenococcus, Vagococcus,* and *Weissella*.

The focus was to review the in vivo scientific and clinical evidence for the administration of probiotics and to further assess probiotic formulations comprising *S. thermophilus* on the intestinal microbiome that could influence reductions in local (intestinal) uremic toxin levels and, thus, inhibit the progression of CKD states.

## 2. Materials and Methods

The following databases were searched to retrieve journal articles and these included PubMed, the Cochrane Library, Science Direct, Scopus, EMBASE, CNKI, CINAHL, and Google Scholar.

PubMed was searched using the MeSH headings chronic kidney disease or hemodialysis; gut dysbiosis; uremic toxins; uremia; p-cresyl sulphate; indoxyl sulphate, and these terms were then combined with probiotics and prebiotics and synbiotics and *S. thermophilus*. The requisite search was for any type of human clinical trial or laboratory animal study, with the administration of probiotics, prebiotics or synbiotics that investigated formulations that included the probiotic bacterium *Streptococcus thermophilus*. The journal article or abstract was to be written in English for inclusion in this narrative review.

## 3. Intestinal Dysbiosis

The healthy cohort of intestinal microbial phyla can be portrayed through their diversity, stability, and resistance, and the resilience to daily perturbation [4]. Correspondingly, the intestinal microbiome depicted as the richness of the ecosystem, its amenability to perturbation, and its ability to return to the pre-perturbation state is synchronized with health. Intestinal dysbiotic configurations of the gut phyla as can occur with CKD have often been portrayed as disrupting mucosal immunity [10,11].

Dysbiosis, therefore, emerges as an event that describes not only a compositional and functional shift in the intestinal microbiota, but also addresses the set of host-associated and environmental factors that progresses the dysbiotic bacterial state in the intestines toward a continued disequilibrium [6].

An altered ratio of intestinal bacterial genera, as may occur with the reduction of commensal bacterial clades and an increase in pathobionts, are common typical features often referred to in the literature as characteristic of dysbiosis. Hence, dysbiosis addresses aggravating intestinal microbiome adverse shifts, with a consequent encouraged pro-inflammatory effect (Figure 1) that disrupts the intestinal microbiome–epithelia–mucosal immunity axis [12]. Intestinal dysbiosis provides a likely clue as to the origin of systemic metabolic disorders encountered in clinical practice such as chronic kidney disease.

## 4. Uremia and the Intestinal Microbiome

Uremia, defined as a significantly raised concentration in the blood of urea and other nitrogenous waste compounds that are normally eliminated by the kidneys, is intimately associated with CKD [14]. Moreover, CKD is characterized by uremic molecules with different molecular weights that are by-products of intestinal metabolism, that adversely alter and overwhelm cellular functions. Indeed, uremic illness is attributed to the progressive retention of circulating uremic solutes that the healthy kidney struggles to excrete. Epidemiological studies have consistently reported that CKD is inexorably a progressive disease with very much a consistent decrease of glomerular filtration rate that eventually leads to ESKD [15]. The kidney struggles to excrete toxins is most probably centered on the definition of CKD as an insult that damages the kidney and or with a concomitant reduction in the glomerular filtration rate that is lower than 60 mL/min per 1.73 m^2^ [15]. This is exacerbated by the presence of increased urinary albumin excretion for months or as a long-term adverse outcome that, in time, presents with decreases in the estimated glomerular filtration rate with a concomitant increase in CKD stages (Table 1) [6]. Such changes have been reported to be reflected in a decrease in the percentage of protein-bound uremic toxins and a concomitant increase in serum indoxyl sulphate. The features of uremia identified in patients with end-stage-renal disease may be present to a lesser degree in individuals with a glomerular filtration rate that is barely below 50% of the normal rate, which at 30 years of age ranges between 100–120 mL per minute per 1.73 m^2^ of body-surface area.

Contributing unfavorable dietary metabolic factors that disrupt the intestinal microbiome that progresses intestinal microbiome dysbiosis in patients diagnosed with kidney disease include compromised protein digestion and assimilation and decreased consumption of dietary fiber [16]. The frequent use of antibiotics can also cause disruption of the intestinal cohort of bacteria leading to dysbiosis [2].

The contribution of antibiotic treatments decreases the diversity and alters the relative abundances of members of the intestinal bacterial community [2]. Moreover, some patients reports conclude that the patients exhibited incomplete intestinal microbiome recovery post-treatment with antibiotics [2]. Mechanical factors such as slow intestinal transit time can also trigger a dysbiotic gut in long-term dialysis patients, as does excessive oral iron intake [17,18].

It is generally agreed that individuals are provided with a unique cohort of intestinal microbiota that participates in numerous specific functions in the host’s nutrient-directed metabolism, the maintenance of structural integrity of the intestinal epithelial–mucosal lining, immunomodulation and homeostasis, and protection against pathobiont overgrowth [3]. The intestinal microbiota in addition to being composed of different bacterial species that taxonomically have been traditionally classified in an ascending direction by genus, family, order, and phyla comprises three main enterotypes, that have been deemed identifiable by the variation in the levels of one of three genera namely, *Bacteroides* (enterotype 1), *Prevotella* (enterotype 2), and *Ruminococcus* (enterotype 3) [4].

In individuals without CKD or other inflammatory gut problems, bacterial species from the phyla Bacteroidetes and Firmicutes provide approximately 90% or greater of all bacterial species, including abundant bacterial genera such as the *Alistipes* spp., *Bacteroides* spp., *Clostridia* spp., *Dorea* spp., *Eubacterium* spp., *Faecalibacterium* spp., *Lactobacillus* spp., *Prevotella* spp., *Porphyromonas* spp., and *Ruminococcus* spp [21]. In addition, the less abundant phyla are represented by the Actinobacteria (i.e., *Bifidobacteria* spp. and *Collinsella* spp.), Proteobacteria (i.e., *Enterobacteriaceae*, *Sutterella* spp., and *Helicobacter* spp.), Verrucomicrobia (i.e., *Akkermansia* spp.), and methanogenic Archaea [21].

There is significant evidence that the intestinal microbiome is altered in CKD [7,22]. A recent study reported that there were distinct disparities in the abundance of 190 bacterial operational taxonomic units (OTUs) between patients diagnosed with end stage renal disease and control groups [23]. Of significant interest is a report that indicates that the intestinal microbiota can produce potentially harmful metabolites [16]. Hence a dysbiotic gut that presents a variable and unbalanced composition of the gut microbiota strongly progresses the posit that the bacterial cohort contributes to the accumulation of uremic retention solutes that have been observed in patients diagnosed with CKD [16].

Inflammation is a constant element relative to the progress of metabolic disease [24]. Moreover, the same group showed that in a study with rats subjected to a 5/6 nephrectomy showed significant differences in the abundance of bacterial OTUs between control animals and the uremic group [23]. The *Lactobacilli* and *Prevotella* (enterotype 2) genera of bacteria showed the most notable decreased abundances in the uremic group of animals [23]. Research from other groups have shown that the number of facultative anaerobic bacteria including *Enterobacteria* and *Enterococci* were higher in patients treated with maintenance hemodialysis than were detected in the controls. Furthermore, that anaerobic bacteria present in hemodialysis patients had significantly lower abundances particularly in the genera *Bifidobacteria* with a concomitant higher abundance of *Clostridium perfringens* [25].

Studies have proposed that there is an influx of compounds such as urea and other retained toxins into the gut lumen [26]. Consequently, reports conclude that there may be a selection pressure that is being exerted on those bacteria that express the urease (e.g., *Pseudomonas* spp.) urate oxidase (e.g., *Enterobacteria* spp., *Clostridia* spp.) and indole and p-cresyl (e.g., *Enterobacteria* spp.) forming enzymes. Furthermore, bacterial families from the *Lactobacilli* and *Prevotella* (enterotype 2) groups have been shown to elaborate short-chain fatty acid forming enzymes and that their abundance has been reported as reduced in patients diagnosed with end-stage renal disease.

The role of the intestinal microbiome in the progression of CKD has been studied in the context of CKD, that has been characterized by changes in the intestinal microbiota composition [23], the accumulation of microbial-derived metabolites [27], disruption of the intestinal barrier functionality and a progressed and maintained level of chronic inflammation [28]. Consequently, a recent study [29] with a small cohort of patients diagnosed with end-stage kidney disease (ESKD) focused on the role of the intestinal microbiota. The composition of the intestinal microbiota was reported to be diverse among these ESKD patients without revealing a common and distinct microbial signature. The intestinal microbial dynamic’s are such, in complexity, relevant to a chronic disease that potentially underscore the difficulties encountered in attempts to modulate the intestinal microbiota. In order to reduce the levels of gut generated and corresponding blood levels of circulating uremic toxin loads with the requisite being to identify specific prebiotic, probiotic and or synbiotic formulations that may beneficially alter gut-derived uremic toxins [29].

## 5. Inflammation, Immunity, and CKD

A chronic inflammatory profile has been repeatedly reported in the majority of patients with CKD, and as such there is an increased prevalence attending the significant decline of kidney function [30,31]. Elevated and increasing levels of CRP with an increased CKD stage of disease indicates that the inflammatory response is a marker for progression of the disease [5,31]. In the group of patients with middle-to-late stage CKD, systemic and sustained pro-inflammatory profiles are associated with adverse health outcomes that include reduced and poor quality of life issues and increased morbidity and mortality due to the increased risk of cardio-vascular diseases and infectious complications [32]. These adverse outcomes are linked to clinical complications such as immune dysfunction acquisition, depression, osteoporosis, and metabolic and nutritional disorders such as diabetes [33]. Inflammatory biomarkers such as interleukin (IL)-6 has been reported to be perhaps the strongest predictor of comorbidity and progression of CKD [34]. The driving force for this sustained inflammatory response has been posited to begin in the gut [35].

Intestinal and systemic inflammatory processes have been documented to be maintained by the excessive production of uremic toxins that is significantly dependent on a dysbiotic intestinal bacterial cohort [12] that predominantly occupies the colonic lumen. Consequently, emerging data from laboratory animal studies and commentaries confirm the notion [23,36] that intestinal dysbiosis-associated intestinal epithelia disrupted barrier functionality is intricately linked to aberrant local mucosal immunity [37] that progresses systemic inflammation and the continued development of the advanced stages of CKD [10].

Macrophages provide important functional roles in tissue homeostasis and immune responses in normal and diseased kidneys [38]. In the intestinal mucosa, macrophages in their eradication of pathogenic microbes so as to maintain homeostasis, are an essential component of innate immunity [39]. The activated phenotype exists in macrophages that present antigens to T lymphocytes in order to initiate an appropriate immune response after recognition of microbial proteins. In addition to serving as antigen-presenting cells, activated macrophages secrete a range of cytokines such as interleukin (IL-1), interferon (IFN-α), and cytotoxic proteins that activate T cell lymphocytes. Moreover, as part of their overall action, macrophages as for many tissues and including the kidney, can phagocytize exogenous antigens, cellular debris, insoluble particles, and activated clotting factors [40]. Therefore, inflammation maladaptation due to the overload of systemic circulating uremic toxins presents a clinical scenario of sustained inflammatory responses that progresses CKD.

## 6. Probiotics, Prebiotics, Synbiotics, and CKD

A recent extensive review reported on the potential efficacy that probiotics may provide in the treatment of CKD [41]. The intestinal microbiome from patients diagnosed with CKD has been profiled as dysbiotic with disruption of the intestinal epithelial cell barrier that promotes an inflammatory phenotype and its associated uremia. The potential efficacy of probiotics in the management of CKD is thought to be partially based on rescuing a dysbiotic intestinal microbiome that helps reduce the concentration of uremic toxins [41].

In this narrative review we have focused on probiotic and symbiotic formulations containing the probiotic bacterium *S. thermophilus,* given this bacterium’s potential to reduce uremic toxins.

*Streptococcus thermophilus* has been shown to reduce indoxyl sulphate in in vitro studies and in clinical studies. *Streptococcus thermophilus* combinations have demonstrated significant efficacy in CKD as evidenced by reductions in the accumulation of circulating uremic toxins (Table 2).

However, the administration of probiotics or prebiotics or probiotics plus prebiotics in a synbiotic formulation has been recently reported as inconclusive, albeit data reviewed from an acceptable number of RCTs [42]. A small study though has concluded that the administration of probiotics or prebiotics or synbiotics should perhaps be accompanied by dietary changes such as the inclusion of a low protein diet [43]. The combined probiotic + prebiotic with a dietary protocol of a low protein diet demonstrated efficacy in slowing the progression of CKD.

## 7. Discussion

It is now recognized and accepted that the development and progression of CKD is, in part, intimately associated with a decrease in the abundance and diversity of the intestinal cohort of bacteria [9]. It has been proposed that the intestines are a forgotten organ in uremia that leads to CKD [34], with the microbiome project indirectly reinforcing this view [56].

We have previously reported that the administration of probiotics is not a panacea [3,13] benefiting the amelioration of all chronic diseases. However, it is biologically plausible that the administration of probiotics with or without prebiotics could beneficially decrease the inflammatory allostatic load associated with retention of middle molecular weight uremic toxins, and synergistically reducing several inflammatory mediators (e.g., IL-6) [40].

Selecting probiotic bacteria to include in formulations that could positively encourage the intestinal microbiome to improve local dysbiosis whilst reducing a gut derived uremia may have biological plausibility.

The two clinical studies that did not report an efficacious outcome following the administration of probiotics, inducted subjects that were on hemodialysis due to poorly to non-functional kidneys [51,52], indicating that probiotic efficacy may depend on the stage of CKD at inclusion into a clinical study (Table 1).Whereas efficacious studies with probiotics in this narrative review consisted of formulations that included *S. thermophilus* with subjects diagnosed with CKD that were not on hemodialysis. Thus, we built a posit in this review that advances a biologically plausible idea that *S. thermophilus* is a probiotic bacterium with mechanistic characteristics in the gut that in conjunction with prudent dietary practices could significantly reduce circulating uremic toxins and stop the progression of CKD. In this regard, the probiotic *S. thermophilus* has been shown in studies with murine models of CKD to efficiently reduce uremic toxins in acetaminophen induced uremia in rats [54,55].

Clustered regularly interspaced short palindromic repeats (CRISPR) are a distinctive attribute of the genetic make-up of most bacteria and archaea and are reported to be involved in resistance to bacteriophages [57]. Indeed, *S. thermophilus* possesses CRISPR and their associated genes that as such links this gene cluster to an acquired resistance profile against bacteriophages [58]. It is accepted that the intestinal bacterial biomass is regulated by the actions of bacteriophages in the human gut, whereby widespread bacteriophage predation and lysogenic conversions of bacterial populations occurs [59]. Moreover, a report that has shown that bacteriophages potentially eliminate pathogenic intestinal bacteria [60]; perhaps indirectly favoring the survival of *S. thermophilus’* in the gut through resistance to bacteriophages affording this probiotic bacterium the capacity to reduce uremic toxins in the gut. This is important, as we have observed (personal communication, 2019) as have others, is that not all probiotic formulations have been reported to be efficacious in reducing uremic toxin loads in the circulation and or the gut [41].

The studies presented in this narrative review favored probiotic formulations that have included *S. thermophilus* and these reports present significantly consistent data with the probiotic formulation investigated in regard to improving uremic toxin levels as well as quality of life and inflammatory markers. Hence, probiotic formulations may require formulations that encompass specific probiotic strains for reducing uremic toxin loads in the gut [41]. As such, *S. thermophilus* may be such a probiotic that should be included in probiotic formulations targeting the early and mid-stages of CKD and not patients undergoing hemodialysis. It would seem that in order to further investigate the efficacy of probiotics in CKD, patient selection should be carefully assessed. It is implausible that the administration of probiotics could ameliorate uremic toxin allostatic loads and reduce inflammatory markers in subjects that have non-functional kidneys and that are consequently undergoing dialysis treatments, as was noted in the Borges and colleagues’ study [51]. Of course, such impressions require further verification given that most studies administered multiple species of probiotic bacteria.

This limited review tends to suggest that formulations that include *S. thermophilus* with or without the addition of prebiotics may be clinically useful in assisting with the reduction of uremic toxins in the intestines. The studies by Patra and colleagues [54] and Mandal and colleagues [55] with murine models of acetaminophen-induced uremia certainly supports the notion that the probiotic *S. thermophilus* may have specific actions in limiting the production of uremic toxins in the gut. In addition, as such though, the clinical use of probiotics with formulations that include *S. thermophilus* may also benefit from patients that can manage a lifestyle change in regard to adopting prudent nutritional practices, thus assisting to also mitigate intestinal bacterial dysbiosis in the lumen and the intestinal–mucosa complex [1,3,5].

## 8. Conclusions

The administration of synbiotic formulations can be effective in improving the composition and metabolic activities of colonic bacterial communities and immune parameters in patients diagnosed with CKD. We posit that the formulations that include the probiotic *S. thermophilus* could be especially useful in controlling the uremic toxin allostatic load in the intestines of patients with CKD not undergoing dialysis. Furthermore, with prudent dietary recommendations [61] and probiotic administration it may be possible to prevent the progression of CKD from the early stages to the later stages of the disease.

## Figures and Tables

**Figure 1 microorganisms-07-00228-f001:**
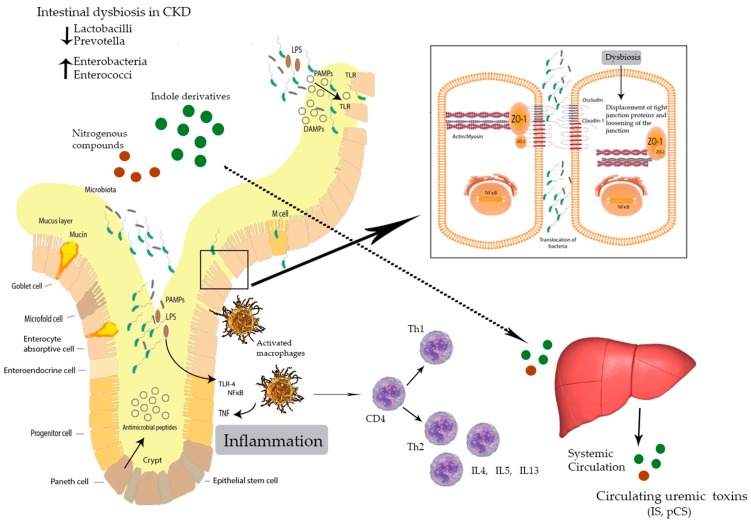
Uremic toxin-induced disruption of the intestinal epithelial tight junctions and intestinally derived uremic toxins progressing uremia [13] (Abbreviations: M cell = microfold cell; TLR-4 = toll Like Receptor-4; NFkB = nuclear factor kappa-light-chain-enhancer of activated B cells; Th1 = T helper 1 cell; Th2 = T helper 2 cell; CD4 = cluster of differentiation 4 cell; IL = Interleukin; LPS = lipopolysaccharide; IS = indoxyl sulphate; pCS = p-cresyl sulphate; PAMPs = pathogen-associated molecular patterns; DAMPs = damage-associated molecular patterns.

**Table 1 microorganisms-07-00228-t001:** Classification and description of the different stages of chronic kidney disease.

CKD Stages	eGFR ^1^ (mL/min/1.73 m^2^)	Report	Mean (SD) % Protein-Bound Uremic Toxins ^1^	Mean (SD) Serum Levels Indoxyl Sulphate ^1^ (µmol/L)
Stage 1	90 mL min^−1^	Normal renal function with abnormal urine report or structural abnormalities or a genetic trait indicating kidney disease.	118 (12)	3.9 (1.1)
Stage 2	60–89 mL min^−1^	Mildly ↓ renal function and other reports (as for Stage 1) indicating kidney disease.		
Stage 3 stage (a)	45–59 mL min^−1^	Moderately ↓ kidney function	111 (11)	6.2 (3.2)
Stage 3 stage (b)	30–44 mL min^−1^			
Stage 4	15–29 mL min^−1^	Severely ↓ kidney function	99 (8)	16.2 (14.9)
Stage 5	<15 mL min^−1^ or patient on dialysis	Very severe or end stage kidney disease (often referred to as established kidney failure)	79 (9)	56.1 (28.6)

^1^ This measurement was taken using the modification of diet in renal disease formula [19] and adapted and modified from Vitetta and Gobe [6]. Mean (SD) percentage protein-bound uremic toxins and mean (SD) serum levels of indoxyl sulphate adapted from Klammt et al., 2012 [20]. Note that the percentage of protein–bound uremic toxins was estimated indirectly based on an estimate of the unbound fraction of a specific albumin bound marker in a sample of plasma [20].

**Table 2 microorganisms-07-00228-t002:** Specific human and laboratory animal interventional studies in Chronic Kidney Disease (CKD) with probiotics, prebiotics, and synbiotic formulations containing *Streptococcus thermophilus*.

Human Studies			
Probiotics Administered	Intervention Details	Results	PubMed ID [Reference]
*Lactobacillus acidophilus* KB31 *Streptococcus thermophilus* KB27 *Bifidobacterium longum* KB35 15 × 10^9^ CFU/day	Single-center, prospective, DBRCT cross-over|*n* = 13|CKD stage 3–4| 6 months	↓ BUN ↓ Uric acid concentration ↑ QoL	PMID 19558344 [44]
*L. acidophilus* KB31 *S. thermophilus* KB27 *B. longum* KB35 15 × 10^9^ CFU/day	Multicenter, prospective, DBRCT cross-over|*n =* 46|CKD stage 3–4| 6 months	↓ BUN ↑ QoL	PMID 20721651 [45]
**Synbiotic:** *L. plantarum*, *L. casei subsp rhamnosus*, *L. gasseri*, *L acidophilus*, *L. salivarius*, *L. sporogenes*, *B. infantis*, *B. longum*, *S. thermophilus and* *previotic inulin (VB Beneo Synergy 1), and resistant tapioca starch*	Single-center|DBRCT cross-over|*n =* 30|CKD non dialyzed stage 3–4|4 weeks	↓ Plasma pCS	PMID 24929795 [46]
*S. thermophilus* KB 19, *L. acidophilus* KB 27 *B. longum* KB 31	Single-center|DBRCT cross-over|*n* = 22|KD| 8 weeks	↑ QoL trend toward ↓ serum IS glucuronide ↓ C-reactive protein	PMID 25147806 [47]
*S. thermophilus* KB 19, *L. acidophilus* KB 27, *B. longum* KB 31 270 × 10^9^ CFU/day	Single-center|*n* = 27| CKD Stage III–IV Dose escalation study from 30 × 10^9^/day to 90 × 10^9^/t.i.d. 4 months	↓ BUN in a subset of plasma samples from 16 subjects	PMID 30774576 [48]
**Synbiotic** *Lactobacillus, Bifidobacteria,* *Streptococcus* genera + prebiotics (inulin fructooligosaccarides galacto-oligosaccarides)	DBRCT|*n* = 37|CKD stage 4–5|6 weeks [4-week] washout|crossover|Dietary advice (protein 0.8 g/kg BW/d)	1° outcomes: level of IS 2° outcomes: levels of pCS; LPS, TMAO, inflammation, and OS markers; RF; QoL ↓Serum pCS only	PMID 26772193 [49]
**Synbiotic:** prebiotic + probiotic	Prospective observation PCRT|*n* = 24|CKD stage 3–4|12 mo|Dietary advice (protein 0.8 g/kg BW/d)	Slowing progression of CKD	PMID 24990390 [43]
*L. casei*, *L. acidophilus*, *L. bugaricus*, *L. rhamnosus*, *B. breve*, *B. longum*, *S. thermophilus*, and fructo-oligosacharide prebiotic 500 mg b.i.d. capsules	RCT|*n* = 66 | CKD stage 3–4|6 weeks	↓ serum urea ↓ serum nitrogen	PMID 27903994 [50]
*L. acidophilus* *S. thermophilus* *B. longum* 90 × 10^9^ CFU/day	Single center|DBRCT| *N* = 46|HD|3 months	No reduction in… → uremic toxins → inflammatory markers	PMID 28888762 [51]
**VSL#3** *S. thermophilus* *B. breve*, *B. longum* *B. infantis*, *L. acidophilus* *L. plantarum*, *L. paracasei* *L. delbrueckii spp bulgaricus* 450 × 10^9^ CFU/sachet/day	Single center|*n* = 16 pediatric patients|HD 3 months	No reduction in… → uremic toxins IS and pCS	PMID 23646054 [52]
**Animal Studies**			
*L. plantarum subsp. plantarum* BCRC12251 *L. paracasei spp. paracasei* BCRC12188 *S. salivarius spp. thermophilus* BCRC13869 at 3 × 10^9^ CFU/kg BW	Rats with cisplatin-induced kidney injury were administered probiotic mix for 5 days.	*↓* Phenol ↓ pCS ↓ IS	No PMID [53]
*S. thermophilus* MTCC1938 At 1 × 10^9^ CFU/mL/100 g BW	Acetaminophen-induced uremic rats were given probiotic for 15 days.	↓ plasma urea ↓ plasma creatinine ↓ urinary protein ↓ urinary glucose ↑ glutathione	No PMID [54]
Commercially available combination formulations at a dose of ≥10^9^ CFU/day **Ecobion:** *L. acidophilus*, *L rhamnosus B. longum*, *B. bifidum* *S. boulardii*, *S. thermophilus* Fructo- oligo-saccharide **VSL#3:** *S. thermophilus* *B. breve*, *B. longum* *B. infantis*, *L. acidophilus* *L. plantarum*, *L. paracasei* *L. delbrueckii spp. bulgaricus*	Acetaminophen-induced uremic rats given one of seven symbiotic combinations for 3 weeks.	VSL#3: *↓* plasma urea |creatinine ↓ glomerular necrosis ↑ CAT, SOD, glutathione	PMID 24740592 [55]

* Abbreviations: PMID = PubMed Identifier; L. = Lactobacillus; B. = Bifidobacterium; S. = Streptococcus; S. = Saccharomyces; B. = Bacillus; C. = Clostridium; PCRT = Placebo Controlled Randomized Trial; DBRCT = Double-Blind Placebo-Controlled Clinical Trial; HD = Hemodialysis; CKD = Chronic Kidney Disease; IS = Indoxyl Sulphate; pCS = p-cresyl sulphate; LPS = Lipopolysaccharides; TMAO = Trimethylamine-N-oxide; OS = Oxidative Stress; RF = Renal Function; BUN = Blood Urea Nitrogen; QoL = Quality of Life; CAT = Catalase; SOD = Superoxide Dismutase; AEs = Adverse Events.

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
