# Peer review of "Gut Dysbiosis and the Intestinal Microbiome: Streptococcus thermophilus a Key Probiotic for Reducing Uremia"

_microorganisms, 2019, doi:10.3390/microorganisms7080228_

Round 1
Reviewer 1 Report
There needs to be a couple of relevant references can be added which may enhance the scientific article.
To add:
1) Metabolic profiling of a CKD cohert reveals metabolic phenotype more likely to benefit from a probiotic. Int J. of pro & pre vol 12 no. 1 pp 43-54, 2017
2) Dose escalation, safety and impart of strain specific probiotic on stages 3 & 4 CKD patients. J. Nephrol ther 2013. 3:141
Author Response
We thank the reviewer for his/her comments.
There needs to be a couple of relevant references can be added which may enhance the scientific article.
We have checked the manuscriopt throughout for syntax and spelling typographical errors.
To add:
1) Metabolic profiling of a CKD cohert reveals metabolic phenotype more likely to benefit from a probiotic. Int J. of pro & pre vol 12 no. 1 pp 43-54, 2017.
ADDED to manuscript
2) Dose escalation, safety and impart of strain specific probiotic on stages 3 & 4 CKD patients. J. Nephrol ther 2013. 3:141
ADDED to manuscript
Reviewer 2 Report
The narrative review entitled: Gut Dysbiosis and the Intestinal Microbiome: Is Streptococcus thermophilus Key to Reducing Uremia? By Vitetta et al aimed to focus on the in vivo scientific and clinical evidence for the administration of probiotics and to further assess probiotic formulations comprising S. thermophilus on the intestinal microbiome influencing uremic toxin levels and thus inhibit the progression of CKD.
-The review does not seem to support the suggested importance of Streptococcus thermophilus in the title. It is not clear to the reviewer why this sole species was selected as the causal organism? Only one study (ref46; to which I had no access to the pdf…) listed in table 2 administered Streptococcus thermophiles as a single probioticum. Why should the effect attributed to the other formulas be attributed to S thermophilus?
-Nowhere the authors made a distinction between the precursors of uremic toxins that are generated in the intestine and the circulating uremic toxins per se. E.g. in figure 1 IS and PCS are depicted in the intestinal lumen while their precursors indole and p-cresol should be depicted there.
-The animal studies apply acetaminophen induced kidney failure. What is the effect of this medication per se since medication is known to have a large explanatory power on microbiome composition.
-Figure 1: what is M cell? At the side of inflammation TLR-4 and NFkB are written does this refer to the macrophages or the intestinal epithelial cells?
-table 2 I would suggest to only focus on human studies, since gut microbiota composition and metabolisms is different from humans and so effects are not necessarily translational. Please group by probiotics and synbiotics. Are there no more details on the symbiotic composition in ref 37? In ref 44 no effects were observed but what parameters were evaluated? What about the effect of Ecobion in ref 47? Isn’t VSL#3 a probioticum?
Minor
-Typos:
Keyword: S. thermophiles
Introduction: omit spaces …psychological/physical…
- Table 1 has no use in this review
-Introduction: the following sentences are not clear:
“An established classification that describes the different stages of chronic kidney disease have been established where the accumulation of gut derived toxins progresses uremic illness.” Do the authors mean that accumulation of gut derived toxins and others is characteristic for CKD progression?
“…uremic toxins such as p-cresyl and indoxyl sulfates and blood bound proteins that are poorly cleared by dialysis,… What do the authors mean by blood bound proteins?
-Why write in past tense under “What this narrative review proposes to add”? at the end the authors write: “…intestinal microbiome that could influence reductions in local (intestinal) uremic toxin levels…” If the authors mean local in the intestine, than these are mainly precursors of uremic toxins.
The following sentence under point 4 (Uremia and the Intestinal Microbiome) is not clear: “Indeed, uremic illness is attributed to the progressive retention of uremic solutes that the healthy kidney struggles to excrete and that in time presents with decreases in the estimated glomerular filtration rate with a concomitant increase in CKD stages. Why does the healthy kidney struggles to excrete?
-please use end-stage kidney disease (ESKD) instead of end-stage renal disease (ESRD)
-line 149: the study of Vaziri is not so recent anymore (2013)
-under 6 line 207 (c.f., Koppe et al 2015).should be omitted.
- sometimes symbiotic is written instead of symbiotic
Author Response
We thank this reviewer for enhancing the manuscript with critical queries.
The narrative review entitled: Gut Dysbiosis and the Intestinal Microbiome: Is Streptococcus thermophilus Key to Reducing Uremia? By Vitetta et al aimed to focus on the in vivo scientific and clinical evidence for the administration of probiotics and to further assess probiotic formulations comprising S. thermophilus on the intestinal microbiome influencing uremic toxin levels and thus inhibit the progression of CKD.
-The review does not seem to support the suggested importance of Streptococcus thermophilus in the title. It is not clear to the reviewer why this sole species was selected as the causal organism? Only one study (ref46; to which I had no access to the pdf…) listed in table 2 administered Streptococcus thermophiles as a single probioticum. Why should the effect attributed to the other formulas be attributed to S thermophilus?
We have amended the title as suggested by the reviewer and further clarified in the discussion the role posited to S thermophilus in a probiotic formulation in CKD.
-Nowhere the authors made a distinction between the precursors of uremic toxins that are generated in the intestine and the circulating uremic toxins per se. E.g. in figure 1 IS and PCS are depicted in the intestinal lumen while their precursors indole and p-cresol should be depicted there.
We have amended Figure 1 accordingly.
-The animal studies apply acetaminophen induced kidney failure. What is the effect of this medication per se since medication is known to have a large explanatory power on microbiome composition.
The animal models with acetaminophen were designed to induce a uremic phenotype in the animals, that is triggering a dysbiotic gut in the animals. We agree many pharmaceuticals will have such an effect on the gut as so happens with antibiotics and PPIs.
-Figure 1: what is M cell? At the side of inflammation TLR-4 and NFkB are written does this refer to the macrophages or the intestinal epithelial cells?
We have added at the end of the Figure a legend of the abbreviations used in the figure.
-table 2 I would suggest to only focus on human studies, since gut microbiota composition and metabolisms is different from humans and so effects are not necessarily translational. Please group by probiotics and synbiotics. Are there no more details on the symbiotic composition in ref 37? In ref 44 no effects were observed but what parameters were evaluated? What about the effect of Ecobion in ref 47? Isn’t VSL#3 a probioticum?
The animal models were included that further supports a biological plausible posit that a damged kidney and induced uremia can ameliorate uremic markers with mulitple or single probiotics that included S. thermophilus.
Minor
-Typos:
Keyword: S. thermophiles
Amended
Introduction: omit spaces …psychological/physical…
Amended
- Table 1 has no use in this review
We have amended Table 1 and we request that it be included demonstrating how CKD progression is associated with a significant increase in uremic toxin levels and a concomitant decrease in protein bound toxins with progressive CKD.
-Introduction: the following sentences are not clear:
“An established classification that describes the different stages of chronic kidney disease have been established where the accumulation of gut derived toxins progresses uremic illness.” Do the authors mean that accumulation of gut derived toxins and others is characteristic for CKD progression?
We have amended the sentence and improved clarity
“…uremic toxins such as p-cresyl and indoxyl sulfates and blood bound proteins that are poorly cleared by dialysis,… What do the authors mean by blood bound proteins?
We have amended and clarified this sentence.
-Why write in past tense under “What this narrative review proposes to add”? at the end the authors write: “…intestinal microbiome that could influence reductions in local (intestinal) uremic toxin levels…” If the authors mean local in the intestine, than these are mainly precursors of uremic toxins.
We have amended the tense
The following sentence under point 4 (Uremia and the Intestinal Microbiome) is not clear: “Indeed, uremic illness is attributed to the progressive retention of uremic solutes that the healthy kidney struggles to excrete and that in time presents with decreases in the estimated glomerular filtration rate with a concomitant increase in CKD stages. Why does the healthy kidney struggles to excrete?
Epidemiological studies have consistently reported that CKD is inexorably a progressive disease with very much a consistent decrease of glomerular filtration rate that eventually leads to ESRD. The why the kidney struggles to excrete toxins is most probably centred on the definition of CKD as an insult that damages the kidney and or with a concomitant reduction in the glomerular filtration rate that is lower than 60 mL/min per 1.73 m2 …this is also believed to be exacerbated by the presence of increased urinary albumin excretion for months or as a long term adverse outcome.
-please use end-stage kidney disease (ESKD) instead of end-stage renal disease (ESRD)
Amended
-line 149: the study of Vaziri is not so recent anymore (2013)
Included but ‘recent’ has been omitted
-under 6 line 207 (c.f., Koppe et al 2015) should be omitted.
Omitted
- sometimes symbiotic is written instead of symbiotic
Amended
Round 2
Reviewer 2 Report
The narrative review entitled: Gut Dysbiosis and the Intestinal Microbiome: Streptococcus thermophilus a Key probiotics for Reducing Uremia? By Vitetta et al was revised and has improved.
comments
-I still would prefer the use end-stage kidney disease (ESKD) instead of end-stage renal disease (ESRD)
-Table 1:
Unit of eGFR should be added to the title of the column being ml/min/1.73m2
The protein binding data added from a publication by Klammt et al are somewhat confusing. According to the title in the table the reader expects the percentage of protein binding (of IS), which cannot be more than 100%, however what Klammt presents is the albumin binding capacity of a sample, which is not explained in the current review.
In the footnote of the table median values are referred while in the table titles mean are presented, please clarify.
-Some typos remain in the text
1) line 132: indoxyl sulfate instead of indoxyl sulphate
2)line 211: maintained
3)line 212: kidney
4) in the table PCS is written in abbreviated using all capital letter, while in the figure “pCS” is used, please make uniform
5) line 348: uremia
6) line 353-354: “In conjunction with this report it has been recently shown that bacteriophages potentially eliminate pathogenic intestinal bacteria [57] that not surprisingly S. thermophilus benefits from resistance to bacteriophage infections.” Do the authors mean that S. thermophilus is a pathogenic bacterium? Is there evidence that bacteriophage would eliminate S. thermophilus?
7) line 369: administered
Author Response
We again thank the reviewer for the comments that continue to enhance the manuscript.
We provide the following answers to the queries received.
The narrative review entitled: Gut Dysbiosis and the Intestinal Microbiome: Str coccus thermophilus a Key probiotics for Reducing Uremia? By Vitetta et al was revised and has improved.
comments
-I still would prefer the use end-stage kidney disease (ESKD) instead of end-stage renal disease (ESRD)
We agree and have amended, apologies for the oversight.
-Table 1:
Unit of eGFR should be added to the title of the column being ml/min/1.73m2
Amended
The protein binding data added from a publication by Klammt et al are somewhat confusing. According to the title in the table the reader expects the percentage of protein binding (of IS), which cannot be more than 100%, however what Klammt presents is the albumin binding capacity of a sample, which is not explained in the current review.
We have clarified this issue
In the footnote of the table median values are referred while in the table titles mean are presented, please clarify.
Rectified
-Some typos remain in the text
1) line 132: indoxyl sulfate instead of indoxyl sulphate
The primary language of all authors is English. We have used the American spelling for sulfate given that we were under the impression that the journal preferred that version, however we are more than happy to use the English version of the spelling and as such we have amended the manuscript accordingly.
2)line 211: maintained
Amended
3)line 212: kidney
Amended
4) in the table PCS is written in abbreviated using all capital letter, while in the figure “pCS” is used, please make uniform
Amended
5) line 348: uremia
Amended
6) line 353-354: “In conjunction with this report it has been recently shown that bacteriophages potentially eliminate pathogenic intestinal bacteria [57] that not surprisingly S. thermophilus benefits from resistance to bacteriophage infections.” Do the authors mean that S. thermophilus is a pathogenic bacterium? Is there evidence that bacteriophage would eliminate S. thermophilus?
We do not or have ever entertained the thought that S. thermophilus is a pathogen. Our understanding is that enteric viruses can either lyase or parasitise bacteria in the gut resulting in an important appreciation of phage predation on microbial dynamics. The fact that S. thermophilus has been shown to evade bacteriophage predation may be an important factor in this probiotic’s effectiveness in reducing gut generated uremic toxins. We have further clarified the sentence in the text.
7) line 369: administered
Amended